# Glycation Leads to Increased Invasion of Glioblastoma Cells

**DOI:** 10.3390/cells12091219

**Published:** 2023-04-23

**Authors:** Paola Schildhauer, Philipp Selke, Christian Scheller, Christian Strauss, Rüdiger Horstkorte, Sandra Leisz, Maximilian Scheer

**Affiliations:** 1Department of Neurosurgery, Medical Faculty, Martin-Luther-University Halle-Wittenberg, Ernst-Grube-Str. 40, 06120 Halle (Saale), Germany; 2Institute for Physiological Chemistry, Medical Faculty, Martin-Luther-University Halle-Wittenberg, 06114 Halle (Saale), Germany

**Keywords:** glycation, invasion, glioblastoma, glioma, astrocytes, methylglyoxal, advanced glycation end-products

## Abstract

Glioblastoma (GBM) is a highly aggressive and invasive brain tumor with a poor prognosis despite extensive treatment. The switch to aerobic glycolysis, known as the Warburg effect, in cancer cells leads to an increased production of methylglyoxal (MGO), a potent glycation agent with pro-tumorigenic characteristics. MGO non-enzymatically reacts with proteins, DNA, and lipids, leading to alterations in the signaling pathways, genomic instability, and cellular dysfunction. In this study, we investigated the impact of MGO on the LN229 and U251 (WHO grade IV, GBM) cell lines and the U343 (WHO grade III) glioma cell line, along with primary human astrocytes (hA). The results showed that increasing concentrations of MGO led to glycation, the accumulation of advanced glycation end-products, and decreasing cell viability in all cell lines. The invasiveness of the GBM cell lines increased under the influence of physiological MGO concentrations (0.3 mmol/L), resulting in a more aggressive phenotype, whereas glycation decreased the invasion potential of hA. In addition, glycation had differential effects on the ECM components that are involved in the invasion progress, upregulating TGFβ, brevican, and tenascin C in the GBM cell lines LN229 and U251. These findings highlight the importance of further studies on the prevention of glycation through MGO scavengers or glyoxalase 1 activators as a potential therapeutic strategy against glioma and GBM.

## 1. Introduction

Glioblastoma (GBM, WHO grade IV glioma) is the most common and aggressive astrocytic brain tumor in adults with a high recurrence and mortality. Despite extensive treatment, including surgical resection, radiotherapy, and temozolomide chemotherapy, the median survival for patients diagnosed with GBM is 12–20 months [1]. The invasive nature of GMB leads to cells infiltrating diffusely into the brain parenchyma, making complete surgical resection difficult and promoting recurrence. For GBM cells to infiltrate and disseminate within a tumor, key changes in the energy metabolism, cell adhesion, and remodeling of the extracellular matrix (ECM) are required [2].

The ECM is a complex network of proteins and components, such as laminin, collagen, and proteoglycans, which provide anchorage of the cells and shape the consistency of the tissue [3]. Several ECM molecules involved in migration and invasion are proteoglycans (versican, brevican, cadherins) and glycoproteins (CD44, tenascin C, fibrinogen), which were found upregulated in higher grade gliomas [4]. GBM cells are known to secrete matrix metalloproteinase (MMP) to degrade the ECM, penetrating the surrounding parenchyma [5,6]. Moreover, GBM cells increase their invasiveness by upregulating tenascin C and brevican, thus creating a migration-promoting environment [7,8]. Through the upregulation of integrin receptors, GBM cells are able to bind other ECM molecules, which facilitates migration [9]. Another mechanism that enhances migratory capacities is the epithelial-mesenchymal transition (EMT). GBM cells undergoing EMT lose epithelial characteristics and become more spindle-shaped and motile, with a downregulation of epithelial proteins such as E-cadherin and an upregulation of mesenchymal proteins such as N-cadherin and vimentin [10,11].

As is known for many cancer cells, GBM reprograms their metabolism to gain energy through aerobic glycolysis (Warburg effect) [12]. Due to the inefficient means of generating adenosine triphosphate (ATP) this way, the aerobic glycolysis occurs 10 to 100 times faster in cancer cells [13]. This produces an increased amount of by-products, which are favorable for tumor growth and progression [14]. During glycolysis, 0.1–0.4% of glucose are turned into methylglyoxal (MGO), a regular by-product, through the non-enzymatic elimination of the phosphate group of glyceraldehyde-3-phosphate [15].

MGO is a reactive dicarbonyl and one of the most potent glycation agents known to cause vascular complications of diabetes (neuropathy, retinopathy, nephropathy, and atherosclerosis) and central nervous system disorders [15]. Being 20,000 times more reactive than glucose, MGO reacts with the amino acids lysine, cysteine, and arginine to form advanced glycation end products (AGEs) [13]. This non-enzymatic reaction between the carbonyl groups of dicarbonyls (MGO or glyoxal) or sugars (glucose, fructose) with amino groups of proteins, lipids, and DNA is called glycation. The process of glycation affects all proteins and can cause protein crosslinking, which alters tertiary structures and protein functions [16,17]. In total, 90–99% of MGO is bound to macromolecules and the cellular concentration can reach up to 0.3 mmol/L [18]. Elevated MGO and AGE levels were found to be associated with Alzheimer’s and cardiovascular disease and mortality in type 1 and 2 diabetes [19,20]. AGEs function through various transmembrane receptors inducing oxidative stress, inflammation, dysregulation of signaling pathways, and genomic instability, which can trigger the initiation and progression of cancer [21]. The activation of the receptor for AGEs (RAGE) can, for example, trigger the JNK/AP1 signaling pathway, which promotes cell survival, invasion, and metastasis [21]. In breast cancer tumors, an accumulation of MGO adducts have been found and studies have shown that MGO induces the remodeling of the ECM and the activation of migratory-signaling pathways, enhancing metastatic dissemination [22].

Our preliminary work showed that MGO led to glycation and increased invasion in benign meningioma cells [23]. Similar results were obtained after the glycation of neuroblastoma cells [24]. Here, an increase in cell migration and invasion associated with a reduction in adhesion was detected.

In this study, we analyzed the effect of MGO on the glioma (WHO grade III and IV) cell lines compared to normal human astrocytes (hAs). We focused on the effect of glycation on chemotaxis, adhesion, and invasion. Our results showed that glycation led to an increase in invasion in the GBM cell lines and a decrease in the hA. In addition, we analyzed the effect of glycation on ECM proteins and their potential role in the observed increased invasion.

## 2. Materials and Methods

### 2.1. Cell Lines and Cultivation

The human glioma cell lines U343, U251, and LN229 have been kindly provided by Jacqueline Kessler (Department of Radiotherapy, Martin Luther University Halle-Wittenberg, Halle (Saale), Germany). All three of the cell lines were cultured in RPMI 1640 (Gibco, Thermo Fisher Scientific, Waltham, MA, USA) supplemented with 1% Penicillin-Streptomycin (10,000 U/mL Penicillin/10,000 µg/mL Streptomycin) (Gibco, Thermo Fisher Scientific, Waltham, MA, USA) and 10% fetal bovine serum (FBS, Gibco, Thermo Fisher Scientific, Waltham, MA, USA) at 37 °C in a 5% CO_2_ incubator. The hAs were obtained from ScienCell Research Laboratories (Carlsbad, CA, USA) and cultured with astrocyte media (ScienCell Research Laboratories, Carlsbad, CA, USA)), as recommended by the manufacturer. All plates for hA were coated prior to use with a poly-L-Lysine solution (0.01%, EMD Millipore Corporation, Burlington, VT, USA).

### 2.2. MGO Treatment

The cell lines were seeded and incubated at 37 °C and 5% CO_2_ in an incubator. After 4 h of attachment, the cells were treated with different concentrations of MGO (Merck, Sigma-Aldrich, St. Louis, MO, USA) (0.1, 0.3, 0.6, and 1.0 mmol/L), depending on the experiment. After the treatment, the cells were incubated again at 37 °C 5% CO_2_ for 24–96 h. The untreated cells served as the control.

### 2.3. XTT Assay

The cellular metabolic activity of the glycated glioma cell lines LN229, U343, and U251, and the hAs were measured with a XTT assay (Roche, Sigma-Aldrich, St. Louis, MO, USA) as an indicator of cell vitality. In total, 5 × 10^4^ cells were seeded in 96-well plates (Techno Plastic Products, TPP, Trasadingen, Switzerland) in 100 µL, incubated, and treated with MGO. As a control, a cell-free media without MGO was used. The XTT assay was performed after 24, 48, 72, and 96 h of MGO treatment. After each incubation period, 50 µL of the XTT labelling mixture was added to each well, according to the kit’s instructions. The plate was incubated for 4 h in a humidified atmosphere, 37 °C, 5.0% CO_2_ and then measured at a wavelength of 492 nm using the Tecan Infinite F200 Pro. (Tecan, Männedorf, Switzerland). The XTT assay was also performed using media with only 1% FBS.

### 2.4. Cell Microscopy

In total, 5 × 10^4^ cells were seeded in 24-well plates (TPP) and treated with different MGO concentrations as described above. Microscope imaging was taken 24 and 48 h after treatment. The cells were stained with a propidium iodide solution (PI, 1.0 mg/mL, Sigma-Aldrich) and NucBlue Live Cell Stain ReadyProbes reagent (Thermo Fisher Scientific). The cells were washed with Dulbecco’s phosphate-buffered saline (DPBS, Gibco, Thermo Fisher Scientific) and covered with FluoroBriteTM DMEM (Gibco, Thermo Fisher Scientific) and imaged with a Keyence BZ-800E microscope (Keyence, Neu-Isenburg, Germany). The quantification of PI and DAPI stained cells was performed using the IndentifyPrimaryObjects function of the software CellProfiler (Version 4.2.4, Broad Institute, Cambridge, MA, USA).

### 2.5. Glycation and Immunoblotting

The cells were seeded in 100 mm × 21 mm petri dishes (TPP) and treated with different concentrations of MGO accordingly. After 24 h, the cells were washed twice with ice cold PBS (Thermo Fisher Scientific) and harvested with PBS containing one diluted Pierce™ Protease Inhibitor Mini Tablet EDTA-free (Thermo Fisher Scientific). Benzonase Nuclease (Merck, Sigma Aldrich) was added to cleave the nucleic acid bonds. The proteins were extracted with 1× LDS sample buffer (Invitrogen, Thermo Fisher Scientific, Waltham, MA USA) and heated at 70 °C for 10 min. The protein concentration measurement was performed using the Pierce BCA Protein Assay Kit (Thermo Fisher Scientific) according to the manufacturer’s instructions. Furthermore, 5% β-mercaptoethanol (Carl Roth, Karlsruhe, Germany) and 1× LDS sample buffer was added to the proteins and afterwards heated at 70 °C for 10 min.

The proteins were separated by sodium dodecyl sulphate polyacrylamide gel electrophoresis (SDS-PAGE) using the NuPAGE™ 4–12%, Bis-Tris, 1.5 mm, Mini-Protein-Gels and NuPAGE™ MES SDS Running Buffer (both Thermo Fisher Scientific). Blotting was performed using the iBlot 2 Dry Blotting System (Thermo Fisher Scientific) with iBlot™ 2 NC Regular Stacks (Thermo Fisher Scientific) followed by Ponceau S staining (0.1% Ponceau S, 3% trichloroacetic acid, and 3% sulfosalicylic acid).

The membranes were blocked with 5% skim milk powder (Carl Roth) in TRIS-buffered saline with 0.1% Tween (TBS-T, Sigma-Aldrich). The primary antibodies were added (Table 1) overnight at 4 °C and, after washing with TBS-T 5 times, the secondary antibodies were added for 60 min at room temperature. The protein level of glyceraldehyde 3-phosphate dehydrogenase (GAPDH) was used as the loading control.

The membranes were developed using the SuperSignal West Femto Chemiluminescent Substrate (Thermo Fisher Scientific) and signals were detected with a CCD camera (ImageQuant LAS4000, GE Healthcare, Freiburg, Germany). The quantification of the band intensity was performed using the ImageQuant TL software version 3.0 (GE Healthcare, Freiburg, Germany) and normalized to the corresponding GAPDH bands. All of the bands identified by the CML antibody, which indicate glycated protein, were included in the quantification process.

### 2.6. mRNA Isolation and qPCR

In total, 5 × 10^5^ cells were seeded in 6-well plates (TPP) and treated with 0, 0.3, and 0.6 mmol/L MGO accordingly. After 24 h, the cells were washed twice with ice cold PBS. Afterwards, the cells were harvested in 300 µL of lysis buffer LBP (MACHEREY-NAGEL, Düren, Germany) and the lysate was stored at −20 °C. RNA was isolated using the NucleoSpin^®^ RNA Plus Kit (MACHEREY-NAGEL, Düren, Germany), according to the manufacturer’s instructions. Using the RevertAid First Strand cDNA Synthesis Kit (Thermo Fisher Scientific) 2 µg of RNA was transcribed into cDNA.

Using the Platinum^®^ SYBR^®^ Green qPCR SuperMix-UDG (Invitrogen, Thermo Fisher Scientific,), 0.5 µL of the respective reverse and forward primers (Table 2) and 1 µL of cDNA were prepared in a total volume of 20 µL. qPCR was performed with the Rotor-Gene Q (Qiagen, Hilden, Germany).

### 2.7. Real-Time Cell Analysis

The chemotactic migration was measured using the Real-Time Cell Analyzer Dual Purpose (RTCA DP) Analyzer (ACEA Biosciences Inc., San Diego, CA, USA), along with the cell invasion and migration plate (CIM-plate 16, ACEA Biosciences Inc.).

In total, 160 µL of media with 20% FBS was added to the lower chamber and 50 µL of media containing 1% FBS was added to the upper chamber. The CIM-plates were assembled according to the manufacturer’s instructions and incubated for 1 h at 37 °C, followed by background measurements. The cells were detached using Accutase (Capricorn Scientific GmbH, Ebsdorfergrund, Germany) and resuspended with 1% FBS media. In total, 2 × 10^4^ cells in 100 µL were added to each well. After 30 min of incubation at room temperature, 0.3 mmol/L or 0.6 mmol/L MGO were added. The cell migration was measured with the RTCA DP Analyzer as a change in impedance every 15 min for 48 h and displayed with the RTCA program 2.0 (ACEA Biosciences Inc.). To analyze the invasion, the upper chamber of the CIM-plate was coated with 20 µL of Geltrex TM LDEV-FREE Reduced Growth Factor Basement Membrane Matrix (Thermo Fisher Scientific). The Geltrex Matrix solution gels were kept at 37 °C, forming a basement membrane and acting as a barrier through which the cells have to invade. The coated upper chambers were incubated for one hour at 37 °C for the Geltrex to polymerize. Afterwards, the CIM-plates were assembled and measured as described above. 

For the adhesion assay, 96× E-plates (ACEA Biosciences Inc.) were coated with 10 µg/mL of Fibronectin (EMD Millipore Corporation) or collagen IV (collagen from human placenta, Bornstein and Traub Type IV, Sigma Aldrich) and incubated for 1 h at 37 °C. Afterwards, the wells were washed with PBS and incubated with media for 20 min. To acquire the background measurements, 50 µL of media with 1% FBS was added to each well. Furthermore, 2 × 10^4^ cells in 100 µL were added to each well. After 30 min of incubation at room temperature, 0.3 mmol/L or 0.6 mmol/L of MGO were added. Adhesion was measured every 15 min for 24 h with the RTCA.

### 2.8. Statistical Analysis

All analyses were performed using Excel Software (Microsoft Corporation, Redmond, WA, USA) and GraphPad Prism 4.9.1 (GraphPad Software Inc., San Diego, CA, USA). The half maximal inhibitory effect (IC50) was calculated using GraphPad Prisms nonlinear regression analysis and the dose-response inhibition equations.

An unpaired two-sided Student’s *t*-test was performed for all cell lines compared to the untreated cells. The figures depict the mean and standard deviation (SD), respectively. At least three biological replicates were performed for each experiment.

## 3. Results

### 3.1. High MGO Concentrations Lead to Decreased Cell Vitality

The influence of MGO on the cell vitality of LN229, U343, U251, and hA after 24, 48, 72, and 96 h was investigated using an XTT assay (Appendix A). MGO showed concentration-dependent cytotoxic effects in all cell lines after 24 h (Figure 1). In LN229, the cell vitality decreased in a concentration-dependent manner, with a reduction in the cell vitality of 18.53 ± 2.71%; *p* < 0.001with 0.1 mmol/L, 34.15 ± 15.07%; *p* < 0.001 with 0.3 mmol/L, 58.28 ± 14.90%; *p* < 0.001 with 0.6 mmol/L; and the strongest effect with 1 mmol/L with a reduction of 64.53 ± 10.11%; *p* < 0.001 (Figure 1A). Equivalent results were observed in the U251 and U343 cell lines (Figure 1B,C). The treatment with 0.1 mmol/L MGO did not affect the cell vitality of the U251 cells and 0.3 mmol/L MGO showed a 20.54 ± 10.21%; *p* = 0.047 reduction in the cell vitality. The 0.6 mmol/L MGO decreased the cell vitality by 51.13 ± 0.34%; *p* < 0.001 and 1 mmol/L by 57.69 ± 4.00%; *p* < 0.001 (Figure 1B). In the U343 cells, 0.1 mmol/L reduced the cell vitality by 17.43 ± 11.47%; *p* = 0.018, 0.3 mmol/L reduced it by 35.96 ± 10.36%; *p* < 0.001, and 0.6 mmol/L MGO reduced it by 47.86 ± 15.86%; *p* < 0.001. A 53.82 ± 17.02%; *p* < 0.001 reduction in the cell vitality was observed at 1 mmol/L MGO (Figure 1C). In the hA cell line, the cell vitality was not affected by treatment with 0.1 and 0.3 mmol/L MGO, but a 51.78 ± 2.53%; *p* < 0.001 reduction in the cell vitality was observed at concentrations of 0.6 mmol/L and a 53.97 ± 2.05%; *p* < 0.001 reduction was observed with 1 mmol/L MGO (Figure 1D). The IC50 of MGO treatment measured 0.384 ± 0.040 mmol/L in the LN229 and 0.4379 ± 0.037 mmol/L in the U251 cells. For the U343 cells, the IC50 was slightly lower at 0.368 ± 0.099 mmol/L and for the hA cells it was slightly lower at 0.333 ± 0.032 mmol/L MGO.

### 3.2. High MGO Concentration Induce Altered Cell Morphology and Cell Death

In addition, we investigated the influence of MGO on the cell morphology. The cells were cultivated in the absence or in the presence of different concentrations of MGO (0.1–1 mmol/L) for 24 h (Figure 2) and 48 h (Appendix A). LN229 and U251 did not exhibit any changes in their morphology when cultured with MGO up to a concentration of 0.6 mmol/L. However, at 1 mmol/L MGO, there was a reduction in the cell amount and the cells became more spherical (Figure 2A,B). U343 and hA displayed a reduction in cell numbers already at 0.3 mmol/L MGO and changes in the morphology at 0.6 mmol/L MGO, appearing more granular and sporadic (Figure 2C,D). An increase in the number of dead cells, as indicated by PI staining, was observed at both 0.6 mmol/L and 1 mmol/L MGO for hA. The quantification of PI-stained cells showed a higher cell death with increasing concentrations of MGO (Appendix A).

In addition, for the LN229 and U251 cells a higher confluency compared to the U343 and hA cell lines was observed. This difference can be attributed to the higher proliferation rate of the GBM cell lines, such as LN229 and U251, in contrast to the grade III glioma cell line U343, and primary normal hA, which have slower growth rates. Additionally, the hA cells have a distinct morphology characterized by spindle-shaped cells with a larger cell body, which further contributes to their lower confluency.

### 3.3. MGO Treatment Increases Glycation in a Concentration-Dependent Manner

To evaluate the level of glycation with the increasing MGO concentration, immunoblotting was performed. The cells were treated with various concentrations of MGO (0.3, 0.6 and 1 mmol/L), and proteins were extracted and separated using SDS PAGE. Carboxymethyl Lysine antibody was used to verify glycation. An increase in glycation could be observed in all four cell lines in a concentration-dependent manner (Figure 3). In the LN229 cell line, protein glycation increased slightly after treatment with 0.3 and 0.6 mmol/L MGO and a 76.77 ± 24.75%; *p* = 0.011 increase was observed with 1 mmol/L (Figure 3A). Similar results were detected in the U251, where 1 mmol/l led to an increase in glycation of 53.09 ± 30.97%; *p* = 0.072 (Figure 3B). In the U343 cell line, 0.3 mmol/L MGO led to no increase in glycation but glycation increased with 0.6 mmol/L MGO by 34.06 ± 14.85%; *p* = 0.032 and with 1 mmol/L by 63.43 ± 33.89%; *p* = 0.057 (Figure 3C). The hA showed the strongest effect of glycation, with an increase of 72.30 ± 62.41%; *p* = 0.231 with 0.3 mmol/L MGO, 93.80 ± 77.57%; *p* < 0.001 with 0.6 mmol/L; and 152.21 ± 81.69%; *p* = 0.223 with 1 mmol/L (Figure 3D). Interestingly, a distinct pattern of glycated proteins was observed in the cell lines.

### 3.4. Chemotactic Cell Migration after MGO Treatment

As chemotactic cell migration plays a crucial role in the dissemination and progression of tumors, we investigated the effect of MGO on chemotaxis. The LN229 cell line exhibited a decrease in cell migration in response to the MGO treatment in a dose-dependent manner, with a reduction of 26.66 ± 10.92%; *p* = 0.043 observed after 48 h of treatment with 0.6 mmol/L MGO (Figure 4A). The treatment with MGO did not result in changes in the chemotactic migration of U251 cells after 24 and 48 h (Figure 4B). The U343 cell line showed a decrease in chemotactic migration activity after treatment with 0.3 mmol/L MGO for 24 and 48 h (Figure 4C). The hA cell line exhibited a reduction in chemotactic migration activity by 11.26 ± 7.53%; *p* = 0.041 with 0.3 mmol/L after 24 h, but no other alterations in chemotactic migration were observed (Figure 4D). The effect of glycation on the cell motility was additionally analyzed using time-lapse microscopy. No significant changes were observed in the migration after MGO treatment compared to the untreated cells.

### 3.5. MGO Increases Invasion of GBM Cell Lines

Since invasiveness is one of the hallmarks of cancer cells, next we analyzed the influence of glycation on invasion. The invasion of the LN229 cells increased significantly with higher MGO concentrations, showing an increase of 23.06 ± 14.46%; *p* = 0.033 after treatment with 0.3 mmol/L and 45.35 ± 18.24%; *p* = 0.025 after treatment with 0.6 mmol/L MGO after 24 h (Figure 5A). Interestingly, the U251 cells only showed an increase in invasion after treatment with 0.3 mmol/L MGO. The enhancement of invasion with 0.6 mmol/L was not significant compared to the control cells (Figure 5B). The U343 cell line showed increased invasiveness with 0.6 mmol/L MGO. No difference was observed between the treatment with 0.3 mmol/L and the untreated cells (Figure 5C). The invasiveness of the hA decreased after MGO treatment in a concentration-dependent manner, showing a reduction of 21.02 ± 8.37%; *p* = 0.023 with 0.3 mmol/L and 35.75 ± 4.54%; *p* < 0.001 with 0.6 mmol/L MGO after 24 h (Figure 5D).

### 3.6. MGO Has No Effect on the Adhesion of Glioma Cell Lines or hA

Since the invasion was altered after glycation, we analyzed the effect of MGO on adhesion. The cells were seeded on different matrices (without coating, fibronectin, and collagen IV) and treated with MGO (0.3 or 0.6 mmol/L). No significant changes of adhesion were observed after glycation in any of the cell lines (Figure 6). However, differences in adhesion to the different matrices were observed. LN229, U343, and hA adhered best to fibronectin, followed by collagen IV and adhered least to the uncoated plates (Figure 6A,C,D). Surprisingly, U251 showed the least adhesion to collagen IV and no difference was measured between the uncoated plates and the fibronectin coating (Figure 6B).

### 3.7. Glycation Alters the Expression of ECM Components

Since cell-cell adhesion molecules, matrix-degrading enzymes, and various ECM components typically modulate invasion, we analyzed the effect of glycation on: versican, tenascin C, MMP 2, MMP 9, MMP 14, fibulin 3, thrombospondin, integrin β1, integrin α3, integrin α5, brevican, fibronectin, vimentin, TGF-β, and transcription factors slug (SNAI2) and snail (SNAI1) (Figure 7).

The effect of glycation on various ECM components, cell-cell adhesion molecules, and matrix-degrading enzymes was analyzed in the LN299, U251, U343, and hA cell lines using qPCR. The mRNA expression levels of these components were found to be lower expressed in the malignant cell lines compared to hA, with the exception of CD44, SNAI1, and fibulin 3 (Figure 7B). The U251 showed a significantly higher expression of CD44 (3.612 ± 1.397, *p* = 0.041) compared to the hA. SNAI1 and fibulin 3 were significantly higher expressed in LN229 (1.794 ± 0.488, *p* = 0.030; 1.712 ± 0.379, *p* < 0.001) (Figure 7B). 

The effect of glycation on the ECM components varied among the cell lines, with an overall upregulation in the LN229 cells. The strongest effect of glycation was observed on the expression of CD44, which was upregulated in the U251 (1.655 ± 0.259, *p* = 0.023), U343 (1.548 ± 0.562, *p* = 0.240), and hA (1.641 ± 0.657, *p* = 0.058) cells, but remained unchanged in the LN229 cells (1.055 ± 0.145, *p* = 0.539). Brevican expression was upregulated in the LN229 (1.498 ± 0.369, *p* = 0.057) and U251 cells (1.303 ± 0.074, *p* = 0.004) and downregulated in the U343 cells (0.725 ± 0.302, *p* = 0.166). Similar results were observed with tenascin C, as the expression increased in the LN229 (1.203 ± 0.140, *p* = 0.046) and U251 cells (1.655 ± 0.238, *p* = 0.018). In the LN229 cells, versican and thrombospondin were upregulated (1.358 ± 0.174, *p* = 0.012; 1.392 ± 0.151, *p* = 0.021), as well as SNAI1 and SNAI2 (1.335 ± 0.160, *p* = 0.011; 1.491 ± 0.294, *p* = 0.027). The TGF-β expression increased in the LN229 and U251 cells (1.229 ± 0.099, *p* = 0.007; 1.328 ± 0.596, *p* = 0.281). The MMP2 expression decreased after glycation in the U251 cells (0.625 ± 0.094, *p* < 0.001). On the contrary, the MMP2 expression increased in the LN229 and U343 cells after glycation (1.518 ± 0.294, *p* = 0.022; 1.544 ± 0.466, *p* = 0.089). No expression of MMP9 was detected in any of the analyzed cells. The remaining components were not differentially expressed by glycation.

In addition, we examined E- and N-cadherin, as they are involved in the epithelial mesenchymal transition (EMT). Therefore, the cells were treated with MGO and immunoblotting was performed using E- and N-cadherin antibodies (Figure 8).

In the LN229 cell line, the expression of both E- and N-cadherin remained unchanged after MGO treatment (Figure 8A). In the U251 cell line, the E-cadherin expression increased in a concentration-dependent manner, with an increase of 21.74 ± 14.15%; *p* = 0.095, 54.26 ± 26.53%; *p* = 0.044 and 81.86 ± 30.09%; *p* = 0.018 observed with 0.3 mmol/L, 0.6 mmol/L, and 1 mmol/L MGO treatment, respectively. The N-cadherin expression was not affected by MGO treatment (Figure 8B). In the U343 cell line, both the E-cadherin and N-cadherin expressions were altered in a concentration-dependent manner, with E-cadherin increasing and N-cadherin decreasing. The E-cadherin expression increased by about 44.25 ± 22.90; *p* = 0.052 with 1 mmol/L MGO treatment and the N-cadherin expression decreased by about 42.36 ± 7.79%; *p* = 0.002 with 1 mmol/L MGO treatment (Figure 8C). In the hA cell line, the E-cadherin expression was not detected and was not induced after glycation. The N-cadherin expression was reduced in a concentration-dependent manner in the hA cells, with a reduction of 42.27 ± 29.44%; *p* = 0.112 with 0.3 mmol/L, 58.27 ± 20.45; *p* = 0.014 with 0.6 mmol/L; and 83.93 ± 8.11; *p* < 0.001 with 1 mmol/L MGO (Figure 8D).

## 4. Discussion

In earlier studies, MGO was initially thought to be toxic to cancer cells and, thus, considered as a therapeutic agent. However, recent studies have revealed that sub-toxic low doses of MGO can promote tumor development, as cancer cells acquire resistance to apoptosis and enhanced growth properties [25,26,27,28,29,30,31,32]. The glycolytic switch of cancer cells (Warburg effect) and increased glycation could positively impact signaling pathways, promoting tumor invasion and uncontrolled cell proliferation. In our study, we analyzed the effect of MGO on GBM and glioma cell behavior. High MGO concentrations (1 mmol/L) led to cytotoxic effects in all cell lines, but low doses increased invasion in the GBM and glioma cell lines, resulting in a more aggressive phenotype. 

Similar outcomes were observed in human meningioma (BEN-MEN-1, WHO grade I) and neuroblastoma (Kelly) cells, where elevated levels of MGO inhibited cell growth and low levels boosted cell invasion [23,24]. Our findings are supported by Nokin et al. The authors implanted U87MG GBM cells on a chicken chorioallantoic membrane and exposed them to increasing MGO concentrations. Low doses of MGO (0.1 and 0.3 mmol/L) significantly increased the tumor volume compared to the untreated tumors, while higher doses (0.5–3 mmol/L) significantly reduced it [33]. According to Lee et al., high MGO doses reduce the crucial cell survival signaling pathway, gp130/STAT3, leading to an increased cytotoxicity in rat schwannoma RT4 cells, PC12 cells, and U87MG GBM cells. Lee did not observe significant harm to cell viability at a concentration of 0.5 mmol/L of MGO [34].

The pro-tumorigenic effect of MGO has been extensively studied previously [31]. One of the key mechanisms supporting cancer progression is the evasion of programmed cell death and the inhibition of tumor suppressors. This can occur as a result of the glycation of heat-shock proteins; for instance, MGO-modified heat-shock protein 27 prevents apoptosis of cancer cells in lung and gastrointestinal cancer [25,35]. In breast cancer, the MGO-altered heat shock protein 90 decreased the LATS1 expression, a kinase of the Hippo tumor suppressor pathway, enhancing growth and metastatic potential in vivo [27].

Sufficient evidence suggests that MGO increases invasion through various mechanisms. In anaplastic thyroid cancer, MGO promotes migration and EMT through modulation of the TGF-β1/FAK signaling pathway [36]. Additionally, glycation has been linked to activating the RAGE/TLR4/MyD88 signaling pathway and upregulating MMP9 expression in breast cancer, thus increasing migration and invasion [37]. Moreover, MGO adduct accumulation, which is a consistent feature of high-stage colon carcinomas, has been linked to the promotion of proliferation, invasion, and EMT through the PI3K/AKT signaling pathway [38].

However, the regulations by glycation in cancer cells are described as differential. In hepatocellular carcinoma, glycation impaired migration and adhesion [39]. Interestingly, Selke et al. showed that glycation reduced the invasiveness of WHO grade III IOMM-Lee meningioma cells [23]. Our study also observed a decrease in the invasiveness of normal primary human astrocytes. This implies that the effects of glycation are cell-type specific. Furthermore, the effect of glycation on the ECM components can also be differential. While there is strong evidence for an EMT-like process in GBMs through well-described EMT-promoting pathways, such as ZEB1/ZEB2, SNAI1, SNAI2, TWIST, and the WNT-catenin pathway, we did not observe EMT to be the primary reason for increased invasion in our data [11]. Instead, we propose a reverse process of the mesenchymal-epithelial transition (MET) in the WHO grade III glioma cell line U343 (E-cadherin increase and N-cadherin decrease). Interestingly, Selke et al. also found an upregulation of E-cadherin and a downregulation of N-cadherin in meningioma cells (BEN-MEN-1, WHO grade I), in which increased invasion was observed, suggesting MET potentially having a role in the increased invasion [23]. According to Their et al., carcinoma cells sometimes undergo MET after dissemination to distant tissue sites and ensuing extravasation in order to efficiently form metastases [40]. In the GBM cell line LN229, even though SNAI1 and SNAI2 was upregulated, E- and N-cadherin remained unaffected while also showing no EMT promoting expression pattern. 

While E-cadherin expression is commonly described as non-abundant or absent in gliomas and GBM [41], our data showed E-cadherin expression in all three malignant cell lines (LN229, U251, U343) and an absence in the hA. Additionally, glycation even increased the E-cadherin expression in the U343 and U251 cell lines, where invasion was increased. The E-cadherin expression has been found in certain subtypes of glioblastoma with epithelial and pseudo-epithelial differentiation, where E-cadherin levels even correlated with a worse prognosis [42]. In a Xenograft mouse model by Lewis-Tuffin et al., the E-cadherin expression also correlated with the increased invasiveness of glioma cells. Additionally, endogenous E-cadherin expression promoted the growth and migration of the SF767 glioma cell line [42]. Together with these findings, our results could suggest a currently unknown role for E-cadherin in GBM. 

Aberrant N-cadherin expression has been reported in many types of cancer, such as lung-, breast-, prostate-, and squamous cell cancer and has been linked to cell transformation, adhesion, apoptosis, angiogenesis, and invasion [43]. In some studies, N-cadherin expression was found to be upregulated in GBM and linked to the extracellular signal-regulated kinase (ERK) pathway, promoting cancer stem cell invasion [44]. Other researchers reported an inverse correlation between N-cadherin expression and invasion and that the downregulation of N-cadherin was linked to changes in cell polarization and abnormal motile behavior, resulting in increased tumor cell migration and invasiveness [45]. However, no consistent association between N-cadherin and invasiveness has been found in glioblastoma and glioma. We also did not find a correlation between invasiveness and N-cadherin expression. 

Glycation also made cell line specific alterations on other ECM components. In our study, we found that glycation increased the expression of CD44 in the U343, U251, and hA cell lines. CD44 is a transmembrane glycoprotein which binds hyaluronic acid in the ECM and is recognized as a molecular marker for cancer stem cells. The binding of hyaluronic acid activates various signaling pathways, leading to cell proliferation, adhesion, migration, and invasion [46]. In GBM, CD44 has been linked to increasing tumor invasiveness, proliferation, and chemotherapy resistance [47]. However, in our study, CD44 upregulation was not associated with the increased invasion of the GBM cell lines. 

Brevican and tenascin-C were upregulated in the LN229 and U251 cells, where the strongest increase in invasion was observed. Both brevican and tenascin-C are glycoproteins in the human brain that are overexpressed in glioma cells and associated with a later tumor stage [48,49]. Brevican promotes glioma cell motility through the upregulation of integrins and proteolytic cleavage by ADAMTS4 [50]. According to Xia et al., tenascin-C increases the GBM invasion of MMP12 and ADAM9 and negatively regulates proliferation [51]. Although strong evidence suggests that versican and thrombospondin play a role in tumor invasion [4,8,52], our study found that they were only upregulated in the LN229 cells. For example, versican enhances locomotion and reduces cell adhesion of astrocytoma cells through the binding of its G1 domain to hyaluronan [53]. Interestingly, thrombospondin is upregulated upon TFG-β stimulation and enhances microtube formation, which form important structural cell networks of GBM contributing to invasion and treatment resistance [54]. The upregulation of MMP2, which we observed in the LN229 and U343 cells, could increase invasion through remodeling and degradation of ECM.

Despite the differential gene expression of the ECM components, we did not observe any effects of glycation on chemotactic cell migration or adhesion at physiological concentrations. Notably, we only analyzed cell-matrix adhesion and not cell-cell adhesion. Invasion, however, is a fine balance between cell-cell and cell-extracellular matrix adhesions. We concluded that the GBM and glioma cell lined preferred fibronectin as substrate more than collagen IV. Fibronectin is a glycoprotein in the brain parenchyma and its expression is increased in brain malignancies [55]. Collagen IV is less common in the brain and presence is usually restricted to the basement membrane of blood vessels or the glia limitants, which explains the poorer adherence [56].

In summary, our study found that high concentrations of MGO are cytotoxic to cells and sub-toxic levels lead to an increase in glycation, resulting in a more invasive phenotype of GBM cells (Figure 9). The underlying mechanism behind this correlation is not yet well understood and further analyses are necessary. Additionally, deglycation could present potential for novel therapeutic approaches, such as utilizing MGO scavengers or activating glyoxalase 1. In the case of colon cancer, the use of an MGO scavenger, carnosine, has shown promising results by enhancing the efficacy of cetuximab therapy in KRAS-mutated cancer cells [57].

### Limitations

Our study is limited by the fact that we used two glioblastoma cell lines and one glioma cell line. Different cell lines have unique genetic and epigenetic characteristics that affect their responses to glycation. The results may not necessarily apply to primary glioma cells and further analysis is needed to evaluate the effect of glycation in these contexts. Another limiting factor is that the invasion was only analyzed in vitro. The in vitro model does not accurately depict the complexity of the tumor microenvironment, including the impact of immune cell infiltration, physiological parameters (oxygen and pH), growth and angiogenic factors, and the unique composition and stiffness of the ECM.

## 5. Conclusions

In our study, we show that MGO leads to glycation in a concentration-dependent manner. While high concentrations of MGO were cytotoxic, lower concentrations increased the invasiveness of the GBM cell lines. In addition, glycation had differential effects on the ECM components that are involved in the invasion progress, upregulating TGFβ, brevican, and tenascin C in the GBM cell lines.

## Figures and Tables

**Figure 1 cells-12-01219-f001:**
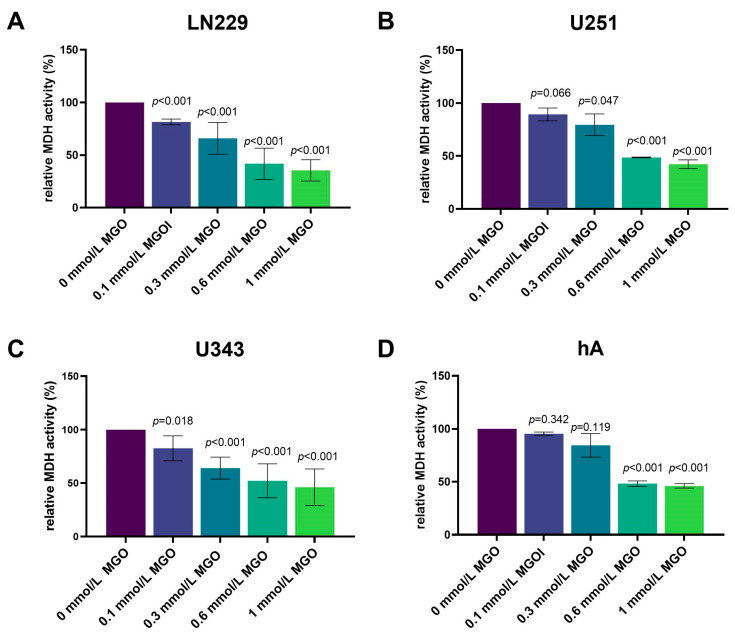
Cell vitality of glioma cell lines and hA after MGO treatment. The cell vitality of LN229 (**A**), U251 (**B**), U343 (**C**), and hA (**D**) cells was determined using an XTT assay after MGO treatment. Graphs show intracellular mitochondrial dehydrogenase (MDH) activity normalized to untreated cells after 24 h. Student’s *t*-test was performed for statistical analysis. Graphs represent the means and SDs of three independent biological replicates.

**Figure 2 cells-12-01219-f002:**
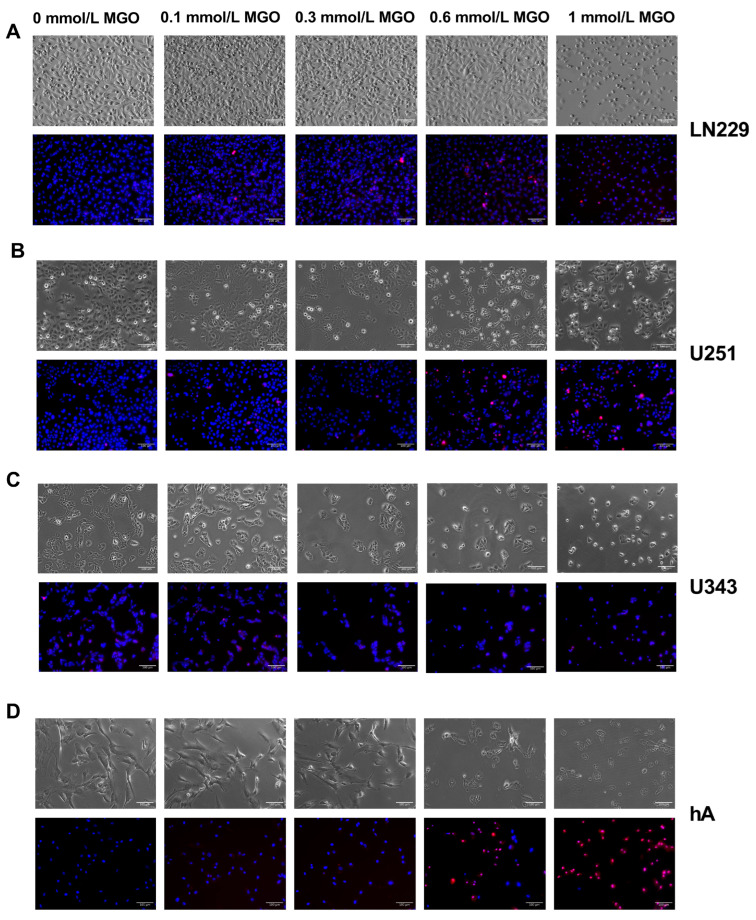
Microscope imaging of glioma cell lines and hA 24 h after MGO treatment. Bright field (above) and fluorescence (below) microscope imaging of LN229 (**A**), U251 (**B**), U343 (**C**), and hA (**D**) 24 h after MGO treatment. Cells were stained with DAPI (blue) and propidium iodide (red). Scale bar = 100 μm.

**Figure 3 cells-12-01219-f003:**
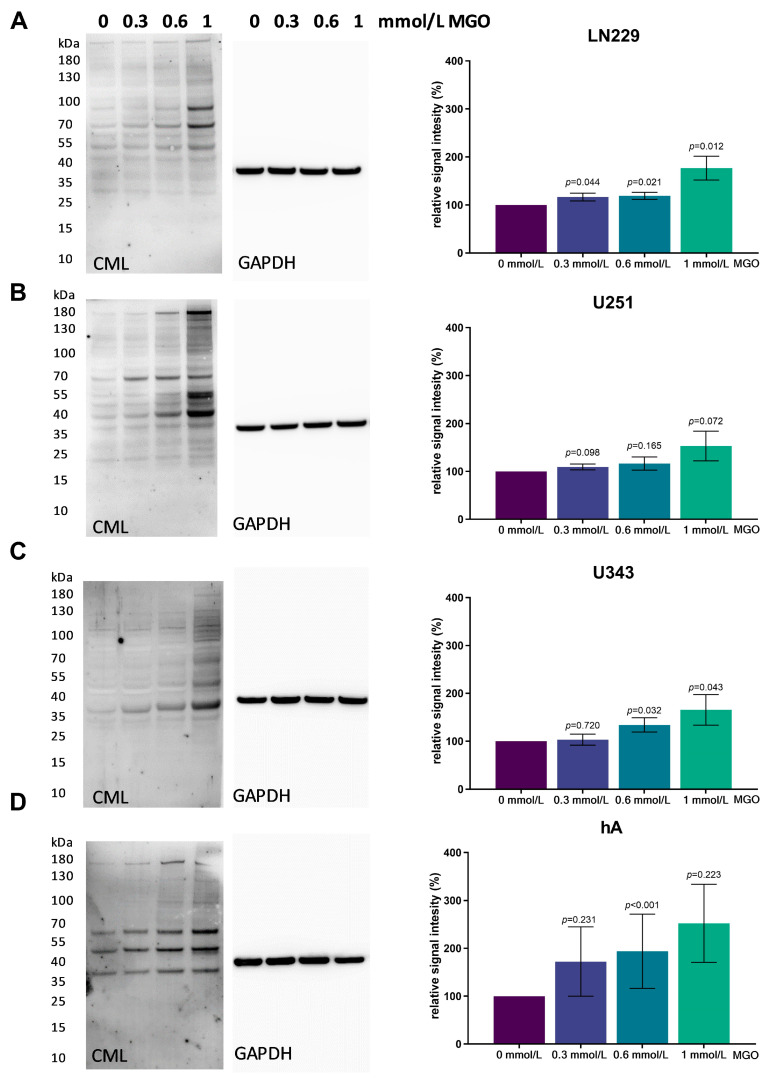
Glycation of glioma cell lines and hA. Immunoblot of LN229 (**A**), U251 (**B**), U343 (**C**), and hA (**D**) with different MGO concentrations (left). Antibody against carboxymethyl lysine (CML) was used to detect glycation. Graphs (right) show representative quantification of the blot, normalized to the untreated cells. GAPDH was used as loading control. Student’s *t*-test was performed for statistical analysis. Graphs represent the means and SDs of three independent biological replicates.

**Figure 4 cells-12-01219-f004:**
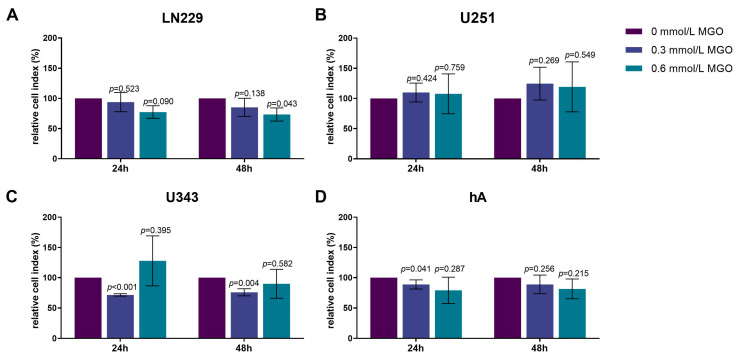
Chemotactic cell migration of glioma cell lines and hA after MGO treatment. Graphs display chemotaxis of LN229 (**A**), U251 (**B**), U343 (**C**), and hA (**D**) after 24 h and 48 h normalized to control cells, after treatment with 0.3 or 0.6 mmol/L MGO. Statistical analysis was performed using Student’s *t*-test. Graphs represent the means and SDs of three independent biological replicates.

**Figure 5 cells-12-01219-f005:**
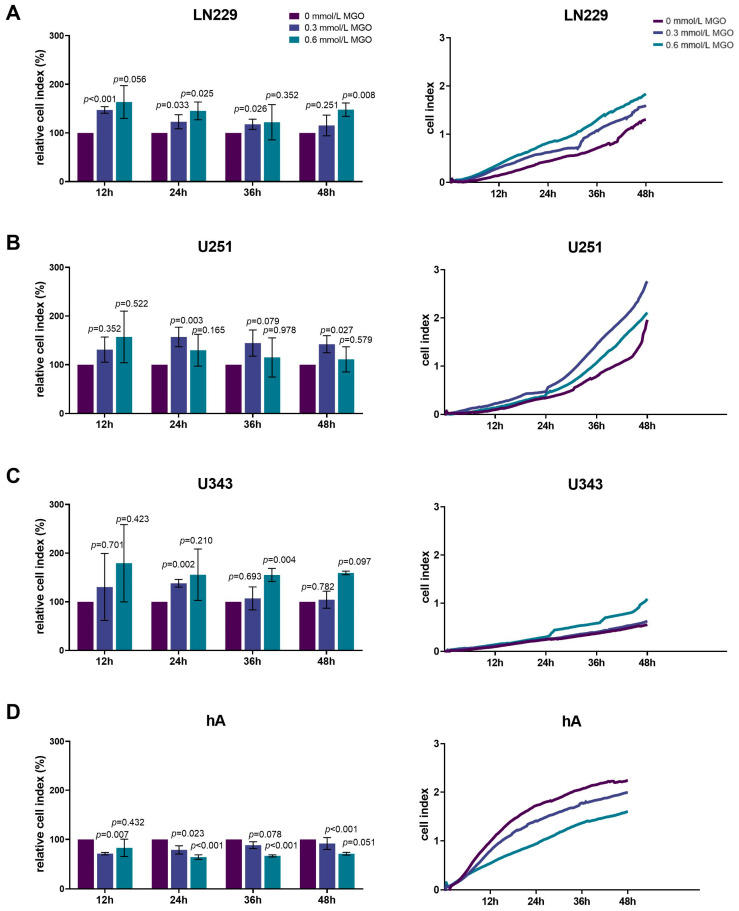
Invasion of glioma cell lines and hA after MGO treatment. LN229 (**A**), U251 (**B**), U343 (**C**), and hA (**D**) were cultivated in absence or presence of MGO (0.3 mmol/L or 0.6 mmol/L) on CIM-plates, coated with Geltrex to imitate basement membranes. Invasion was measured every 15 min for 48 h. Graphs (left column) show cell indices normalized to the untreated cells and graphs (right column) show measured cell indices for 12, 24, 36, and 48 h. Student’s *t*-test was performed for statistical analysis. Graphs represent the means and SDs of three independent biological replicates.

**Figure 6 cells-12-01219-f006:**
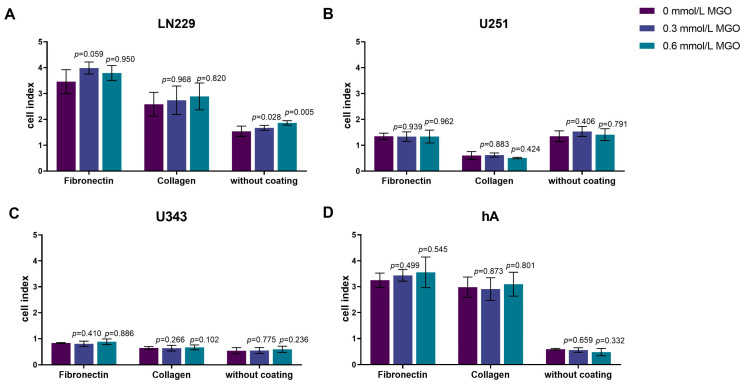
Adhesion of glioma cell lines and hA after MGO treatment. LN229 (**A**), U251 (**B**), U343 (**C**), and hA (**D**) were seeded at concentrations of 0.3 mmol/L and 0.6 mmol/L MGO on plates either coated with fibronectin, collagen, or left uncoated. Graphs display measured cell index after 4 h. Absolute cell index was used to show the different adherence to the different matrices and to illustrate the differences between the cells. Student’s *t*-test was performed for statistical analysis. Graphs represent the means and SDs of three independent biological replicates.

**Figure 7 cells-12-01219-f007:**
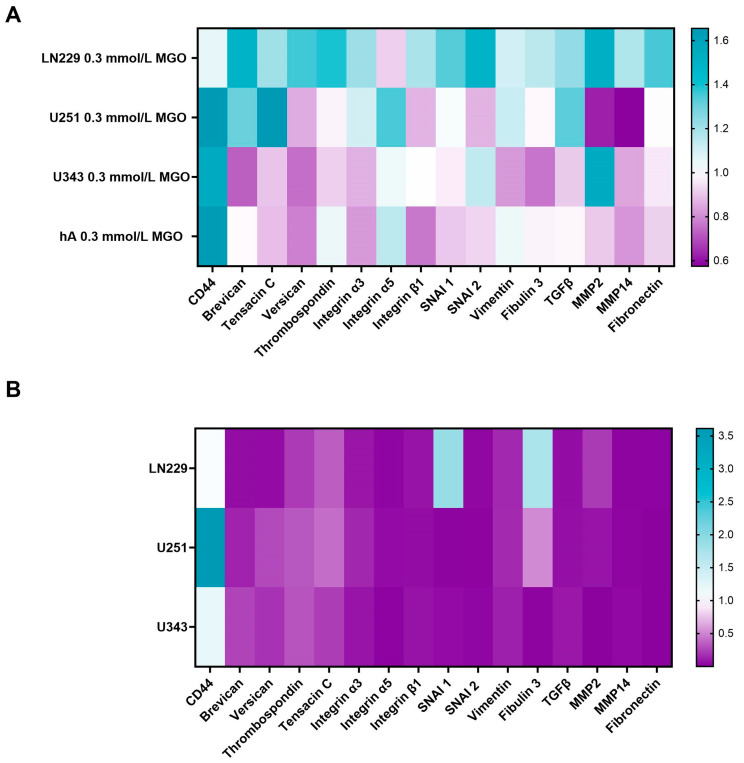
mRNA expression of invasion-associated ECM molecules and transcription factors. Heat map of mRNA expression of LN229, U251, U343, and hA after treatment with 0.3 mmol/L MGO normalized to untreated cells (**A**). Heatmap of mRNA expression of LN229, U251, and U343 cells normalized to the expression of hA (**B**). Three independent biological replicates of the mRNA were analyzed by qPCR.

**Figure 8 cells-12-01219-f008:**
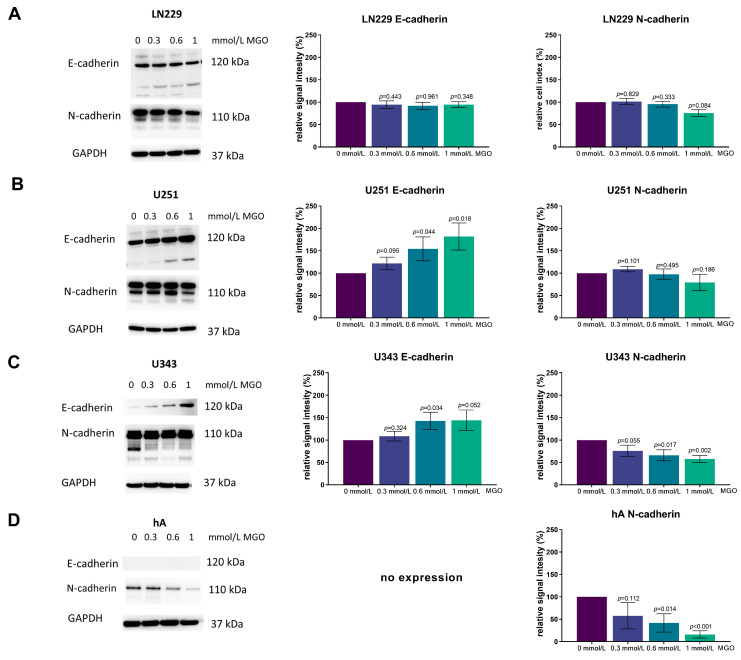
E- and N-cadherin expression after MGO treatment. Immunoblot of LN229 (**A**), U251 (**B**), U343 (**C**), and hA (**D**) cells with different MGO concentrations (0.3, 0.6, and 1 mmol/L) (left column) and antibody against E- and N-cadherin. GAPDH was used as loading control. Graphs show representative quantification of E-cadherin (middle column) and N-cadherin (right column) Western blots from three independent biological replicates, normalized to the untreated cells. Student’s *t*-test was performed for statistical analysis. Graphs represent the means and SDs.

**Figure 9 cells-12-01219-f009:**
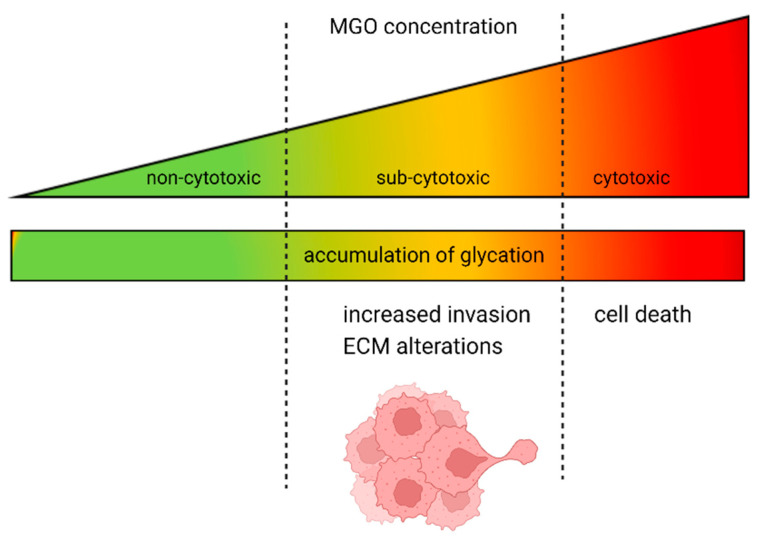
Dual role of MGO in GBM and glioma cells. The increase in glycation is proportional to the rising concentration of MGO. At high doses, MGO has a cytotoxic effect on GBM and glioma cells; however, when present at lower, physiological concentrations, it causes alterations in the ECM and increased invasion.

**Table 1 cells-12-01219-t001:** Antibodies used for immunofluorescence staining.

Antibody	Species	Dilution	Dilution Buffer	Manufacture
Anti-Carboxymethyl Lysine antibody (ab125145)	Mouse IgG	1:1000	5% MP in TBS-T	Abcam (Cambridge, UK)
Anti-E Cadherin antibody Intercellular Junction Marker (ab15148)	Rabbit IgG	1:1000	5% BSA in TBS-T	
Recombinant Anti-N Cadherin antibody (ab245117)	Rabbit IgG	1:1000	5% BSA in TBS-T	
GAPDH (14C10) (#2118)	Rabbit IgG	1:1000	5% BSA in TBS-T	Cell Signaling Technology Inc. (Danvers, MA, USA)
Anti-rabbit IgG, HRP-linked Antibody (#7074)	Goat	1:1000	2% MP in TBS-T	
Anti-mouse IgG, HRP-linked Antibody (#7076)	Horse	1:1000	2% MP in TBS-T	

Abbreviations: BSA, bovine serum albumin; IgG, immunoglobulin G; MP, milk powder.

**Table 2 cells-12-01219-t002:** Primers used for real-time quantitative PCR.

Gene Name (Protein)	Oligo Sequence 5′ to 3′ (Forward, Reverse)	Annealing Temperature (°C)	Product Length	Reference Sequence	Species
CD44	ACGCTTCAGCCTACTGCAAA GGTCCTGCTTTCCTTCGTGT	60	279	NM_000610.4	Homo sapiens
MMP2	ATGTCGCCCCCAAAACGG CCGCATGGTCTCGATGGTAT	60	176	NM_004530.6	Homo sapiens
MMP9	TCTATGGTCCTCGCCCTGAA CATCGTCCACCGGACTCAAA	60	219	NM_004994.3	Homo sapiens
MMP14	GGAGAATTTTGTGCTGCCCG TTGGTTATTCCTCACCCGCC	60	247	NM_004995.4	Homo sapiens
Versican	GCAGAAACTGCATCACCCAG TCCCAGGGCTTCTTGGTACT	60	227	NM_004385.5	Homo sapiens
Brevican	ATGGTGGGACATGCTTGGAG GAAGTCCTGTTCCTCGGGTG	60	233	NM_021948.5	Homo sapiens
Tensacin C	GAAACTGCAGAGACCAGCCT CAGGGGCTTGTTCAGTGGAT	60	244	NM_001410991.1	Homo sapiens
Fibronectin	GGTCCGGGACTCAATCCAAA GACAGAGTTGCCCACGGTAA	60	279	NM_212482.4	Homo sapiens
Integrin β1	AGCAACGGACAGATCTGCAA GCTGGGGTAATTTGTCCCGA	60	241	NM_002211.4	Homo sapiens
Integrin α3	GGCCTGCCAAGCTAATGAGA GACTCACCCATCACTGTCCC	60	273	NM_002204.4	Homo sapiens
Integrin α5	TCTCAGTGGAGTTTTACCGGC CCGAGAGCCTTTGCTGTCAA	60	173	NM_002205.5	Homo sapiens
Fibulin 3	TGTATGTGCCCCCAGGGATA ATTGACTGGGGCAGTTCTCG	60	227	XM_005264205.5	Homo sapiens
Vimentin	GGAGTCCACTGAGTACCGGA AGGTGACGAGCCATTTCCTC	60	198	NM_003380.5	Homo sapiens
Snail (SNAI1)	CTCGAAAGGCCTTCAACTGC GACATTCGGGAGAAGGTCCG	60	298	NM_005985.4	Homo sapiens
Slug (SNAI2)	TTTCAGACCCCCATGCCATT GAAAAAGGCTTCTCCCCCGT	60	292	NM_003068.5	Homo sapiens
Thrombospondin 1	ATCCTGGACTCGCTGTAGGT AGAAAGGCCCGAGTATCCCT	60	209	NM_003246.4	Homo sapiens
GAPDH	TCGTGGAAGGACTCATGACC TTCCCGTTCAGCTCAGGGAT	60	172	NM_002046.7	Homo sapiens

## Data Availability

The dataset is available from the corresponding author upon reasonable request.

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
