# Peer review of "Glycation Leads to Increased Invasion of Glioblastoma Cells"

_cells, 2023, doi:10.3390/cells12091219_

Round 1

Reviewer 1 Report

In this paper, the authors study the invasive brain tumour glioblastoma and how glycation affects glioblastoma cell invasion.

Given the increased interest in glioblastoma cancer, understanding the mechanisms involved in the progression of invasion is very important.

The following are a number of comments for the authors:

Line 101 - authors should control incubation conditions (was incubation performed in an incubator?).

Table 1 - add codes for antibodies to Carboxymethyl Lysine and GAPDH

Line 177 - control the number of cells seeding

3.1, 3.4, 3.5 section - it would be better to add ±SD as in section 3.7 

I suggest the authors calculate IC50 for all cell lines

Figure 1 - why do the authors indicate "24, 48, 72 and 96 hours of MGO treatment" in the methods but only 24 hours are shown in the figure?

Line 227 - the authors could have added cell morphology analysis in the supplementary data.

Figure 2 - the picture of the cells is too small, the authors should enlarge part of the photo to show the cell morphology better.

Figure 2B - for concentrations 0.1 and 0.3 the authors presented the same picture (in blue)

Figure 2C - for concentrations 0 and 0.1 the authors presented the same picture (in blue)

Figure 2 - the authors should explain why they presented different initial confluence of cells for the different cell lines?

Figure 2 - the authors should present a quantitative analysis for the fluorescence images in Figure 2.

Figure 2 - It is known that propidium iodide is commonly used to detect dead cells in a population. How could the authors explain the difference in high levels of red fluorescence in hA cells and low levels in U251, U343 cells which is not in agreement with the cell viability assay?

Figure 2 - why do the authors state "24 and 48 hrs MGO treatment" in the methods but the figure only shows 24 hrs?

2.5 - glycation and immunoblotting - the authors should better explain how they calculate the WB results, especially after the detection of Carboxymethyl Lysine antibody.

Figure 3C - picture of WB does not correspond to the calculated graph

Figure 4 - why didn't the authors treat the cells with 1 mmol/l MGO for the migration assay?

Figure 5 - why didn't the authors add statistical parameters in the right column? 

Figure 5 - the authors describe 'real-time cell invasion' as a method to control cells every 15 minutes, why didn't the authors add time points to the curve to demonstrate a better kinetic curve for invasion of glioma and hA cell lines?

"3.6. MGO has no effect on adhesion of glioma and hA cell lines" - why did the authors not treat cells with 1 mmol/L MGO?

Reviewer 2 Report

In this study, the authors aimed to study the effect of methylglyoxal (MGO) on chemotaxis, adhesion, and invasion of glioblastoma and glioma cell lines compared to normal human astrocytes (hA).  The authors found that MGQ promoted the invasiveness of the GBM cell lines, whereas glycation decreased the invasion potential of hA.  In addition, glycation had differential effects on ECM components that are involved in the invasion progress in the GBM cell lines.  Thus, the authors suggest that further studies on the prevention of glycation through MGO scavengers or glyoxalase 1 activators may be a potential therapeutic strategy against glioma and GBM.   Although this paper presents some interesting data, some points remain to be clarified.  Overall, this study presents observations without in-depth elucidation of the mechanisms leading to these observations.

(1)   Data from this study demonstrated that MGQ exhibited differential effects on the adhesion and invasion of glioblastoma cells, glioma cells, and normal human astrocytes.  The authors should address the mechanisms for the differential effect.  Otherwise, this study provides a superficial observation of MGQ-induced effects.

(2)   The authors suggest that MGO scavengers or glyoxalase 1 activators may be potential therapeutic strategies against glioma and GBM.  It is recommended to examine the effect of MGO scavengers or glyoxalase 1 activators on attenuating MGQ induction to add novelty to this study. 

(3)   Given that MGQ differently induces the production of glycated proteins in the tested cell lines, the authors should investigate the role of glycated proteins in MGQ-induced effects. 

Round 2

Reviewer 1 Report

Authors addressed concerns and modified their manuscript accordingly.

Reviewer 2 Report

Some part of this manuscript has been revised.   However, this manuscript should be revised further.   The authors should analyze the effect of MGO scavengers or glyoxalase 1 activators on attenuating MGQ-induced effects in glioma and GBM cells.  Moreover, the authors should provide some literature to support that glycation significantly increases the malignant progression of glioma and GBM in vivo.